# Heterozygous and Homozygous Variants in *SORL1* Gene in Alzheimer’s Disease Patients: Clinical, Neuroimaging and Neuropathological Findings

**DOI:** 10.3390/ijms23084230

**Published:** 2022-04-11

**Authors:** Maria Isabel Alvarez-Mora, Victor Antonio Blanco-Palmero, Juan Francisco Quesada-Espinosa, Ana Rosa Arteche-Lopez, Sara Llamas-Velasco, Carmen Palma Milla, Jose Miguel Lezana Rosales, Irene Gomez-Manjon, Aurelio Hernandez-Lain, Justino Jimenez Almonacid, Belén Gil-Fournier, Soraya Ramiro-León, Marta González-Sánchez, Alejandro Octavio Herrero-San Martín, David Andrés Pérez-Martínez, Estrella Gómez-Tortosa, Eva Carro, Fernando Bartolomé, Maria Jose Gomez-Rodriguez, María Teresa Sanchez-Calvin, Alberto Villarejo-Galende, Marta Moreno-Garcia

**Affiliations:** 1Genetic Service, Hospital Universitario 12 de Octubre, 28041 Madrid, Spain; juanf.quesada@salud.madrid.org (J.F.Q.-E.); anarosa.arteche@salud.madrid.org (A.R.A.-L.); carmen.palma@salud.madrid.org (C.P.M.); josemiguel.lezana@salud.madrid.org (J.M.L.R.); igomez@salud.madrid.org (I.G.-M.); mariajose.gomezr@salud.madrid.org (M.J.G.-R.); mscalvin@salud.madrid.org (M.T.S.-C.); m.moreno@salud.madrid.org (M.M.-G.); 2Biochemistry and Molecular Genetic Service, Hospital Clínic de Barcelona, 08036 Barcelona, Spain; 3Neurology Department, Hospital Universitario 12 de Octubre, 28041 Madrid, Spain; victorantonio.blanco@salud.madrid.org (V.A.B.-P.); sara.llamas@salud.madrid.org (S.L.-V.); mgonzalezsanchez2@salud.madrid.org (M.G.-S.); alejandrooctavio.herrero@salud.madrid.org (A.O.H.-S.M.); daperezm@salud.madrid.org (D.A.P.-M.); alberto.villarejo@salud.madrid.org (A.V.-G.); 4Network Center for Biomedical Research in Neurodegenerative Diseases (CIBERNED), 28031 Madrid, Spain; carroeva@h12o.es (E.C.); fbartolome.imas12@h12o.es (F.B.); 5Group of Neurodegenerative Diseases, Instituto de Investigación Hospital 12 de Octubre (i+12), 28041 Madrid, Spain; 6UdisGen—Unidad de Dismorfología y Genética, Hospital Universitario 12 de Octubre, 28041 Madrid, Spain; 7Neuropathology Unit, Hospital Universitario 12 de Octubre, 28041 Madrid, Spain; aurelio.hlain@salud.madrid.org (A.H.-L.); justino.jimenez@salud.madrid.org (J.J.A.); 8Genetic Service, Hospital Universitario de Getafe, 28905 Madrid, Spain; belen.gilfournier@salud.madrid.org (B.G.-F.); soraya.ramiroleon@salud.madrid.org (S.R.-L.); 9Neurology Department, Fundación Jiménez Díaz, 28040 Madrid, Spain; egomezt@fjd.es; 10Network Center for Biomedical Research in Cancer (CIBERONC), 28029 Madrid, Spain

**Keywords:** SORL1, Alzheimer, homozygous case, cerebral amyloid angiopathy, SorLA immunohistochemistry

## Abstract

In the last few years, the *SORL1* gene has been strongly implicated in the development of Alzheimer’s disease (AD). We performed whole-exome sequencing on 37 patients with early-onset dementia or family history suggestive of autosomal dominant dementia. Data analysis was based on a custom panel that included 46 genes related to AD and dementia. *SORL1* variants were present in a high proportion of patients with candidate variants (15%, 3/20). We expand the clinical manifestations associated with the *SORL1* gene by reporting detailed clinical and neuroimaging findings of six unrelated patients with AD and *SORL1* mutations. We also present for the first time a patient with the homozygous truncating variant c.364C>T (p.R122*) in *SORL1*, who also had severe cerebral amyloid angiopathy. Furthermore, we report neuropathological findings and immunochemistry assays from one patient with the splicing variant c.4519+5G>A in the *SORL1* gene, in which AD was confirmed by neuropathological examination. Our results highlight the heterogeneity of clinical presentation and familial dementia background of *SORL1*-associated AD and suggest that *SORL1* might be contributing to AD development as a risk factor gene rather than as a major autosomal dominant gene.

## 1. Introduction

Alzheimer’s disease (AD) is a complex multifactorial neurodegenerative disease characterized by progressive cognitive dysfunction and behavioral impairment. Currently, clinical diagnosis of AD remains challenging, and it is based upon clinical presentation fulfilling several criteria as well as fluid and imaging biomarkers [1,2]. Although cerebrospinal fluid and PET biomarkers combined with several relatively new clinical criteria may aid diagnosis in living patients, in most cases, the definitive diagnosis requires post-mortem evaluation of brain tissue [3].

The age of 65 years is used to classify AD patients in two forms: early-onset (EOAD) or late-onset (LOAD). Both show a strong genetic component and are generally considered “polygenic” disorders [4]. Around 95% of AD patients develop LOAD, which is mainly sporadic, with an estimated heritability of 58–79% including the small cumulative effects of multiple variants and genes [5]. Despite advances in genetic diagnosis, only a small portion of the EOAD cases are known to be caused by highly penetrant variants in three major genes, namely, *APP*, *PSEN1* and *PSEN2* [6]. These genes are associated with the rare familial early-onset form of AD with an autosomal dominant transmission and accounts for approximately 5% to 10% of all EOAD (for further review see: [7,8,9]). It has been described that all EOAD mutations in these three genes follow a “reading-frame preservation rule” whereas pathogenic variants truncating the reading-frame do not cause EOAD [10]. However, most EOAD cases are caused by a multifactorial etiology, with the *APOE* ε4 allele being considered the strongest risk factor for AD [11].

In the last few years, increasing evidence has highlighted variants in the *SORL1* gene as risk factors associated with AD [12,13]. The *SORL1* gene is located on chromosome 11q23.2-q24.2. It encodes for a 250 kDa protein named SorLA, which is a functional sorting receptor for the amyloid-β precursor protein (APP). The sorting of proteins is essential for normal cell function, and defects in these pathways are thought to be important factors in the pathogenesis of AD. The functional role of SorLA has been investigated mostly in the complex trafficking pathways of APP within cells [14]. Specifically, SorLA interacts (via N-terminal ectodomain) with APP, retaining it in the trans-Golgi network and reducing its processing into amyloid-β peptide. Mutations in *SORL1* have been shown to reduce the capacity of SorLA to bind APP, resulting in increased levels of sAβPP [15,16]. In addition, it has been recently demonstrated that loss of *SORL1* induces early endosome pathology in neurons and other cell types affected in AD using hiPSC and CRISPR-Cas9 technology [17].

Rare missense and loss-of-function (LOF) variants in the *SORL1* gene have been suggested to show strong association with both the common sporadic LOAD form and the rare familial EOAD [14,18]. It is estimated that rare *SORL1* variants are associated with a five-fold increased risk for EOAD [19,20]. LOF *SORL1* variants are present at an odds ratio between 4, compared to population databases [12], to 12.3 for all AD and 27.5 for EOAD [13]. In fact, the effect on AD risk of *SORL1* variants is currently considered comparable to that of carrying the *APOE* ε4 allele [18]. In this study, we expand the clinical manifestations associated with the *SORL1* gene by reporting detailed clinical and neuroimaging findings of six unrelated Spanish patients with AD and different *SORL1* mutations identified by next generation sequencing (NGS), including for the first time an early-onset dementia patient with a nonsense variant in homozygosis. Furthermore, we report neuropathological findings and immunochemistry assays from one EOAD patient with a splicing variant of the *SORL1* gene.

NGS identified seven patients with candidate variants in dementia causal genes (Appendix A). From the remaining, 13 were found to carry risk factor variants in AD susceptibility genes. These included: nine patients with *APOE* ε4 alleles (one homozygous and eight heterozygous), two with nonsense variants in *SORL1* gene (one in homozygosis (case 1) and one in heterozygosis (case 2)), one patient with a heterozygous rare missense variant in the *SORL1* gene who was also heterozygous for the *APOE* ε4 allele and carried a rare variant in *ABCA7* (case 3), and one patient with the risk variant p.Arg47His in the *TREM2* gene. 

In our cohort, *SORL1* (NM_003105.6) variants were present in 15% (3/20) of individuals carrying either causative or risk factor variants. This high percentage led us to retrospectively review patients with *SORL1* variants from the memory clinic of the Hospital 12 de Octubre. Three of them (cases 4, 5, and 6) were identified to carry potential pathogenic heterozygous variants in the *SORL1* gene, including one splicing variant (this patient has already been reported in [21]) and two missense variants. These variants were identified through NGS studies performed in external laboratories, and patients were referred to the Genetic Service of the Hospital 12 de Octubre, Madrid, Spain, for genetic counseling.

Relevant clinical features of the six patients with *SORL1* variants are summarized in Table 1. Patients with *SORL1* mutations presented considerable heterogeneous phenotypes. Cases 1, 2, 3, and 5 displayed a predominant memory impairment over other domains affected. On the contrary, case 4 showed important behavioral symptoms since the beginning, and case 6 presented with a logopenic aphasia syndrome. In addition, case 2 reported a family history of ischemic stroke and displayed mild small vessel disease on magnetic resonance imaging (MRI) that might have been influenced by *SORL1*; and case 1, as discussed below, had concomitant cerebral amyloid angiopathy (CAA). All of them (except case 3, who was diagnosed with amnestic mild cognitive impairment) met criteria for probable AD following NIA-AA 2011 criteria, and the pathological deposition of amyloid-β was demonstrated in all four patients who were tested. This work may highlight the complexity of *SORL1* contribution to AD.

A molecular description of *SORL1* variants is shown in Figure 1 and Table 2. Variants detected in cases 1, 2, and 4 were considered as pathogenic/likely pathogenic variants, since all are null variants not present in public databases (c.4408C>T, p.R1470* and c.4519+5G>A) or present with an extremely low allelic frequency (c.364C>T, p.R122*) (Table 2). On the other hand, those missense variants (p.S636T, p.G2000R, p.H2038D) detected in cases 3, 5 and 6 were considered as variants of unknown significance (VUS) due to its moderate impact on the protein, their low frequency in the general population, and their in silico pathogenicity predictions (Table 2). However, the family history of the patients herein reported suggest that the *SORL1* gene is contributing to early-onset dementia and specifically to AD as a risk factor gene rather than a causal gene such as *APP*, *PSEN1*, or *PSEN2*.

## 2. Results

### 2.1. Case 1

Patient 1 was a 62-year-old right-handed man who complained of a 3-year history of apathy, memory deficits and spatial difficulties. He was born from non-consanguineous parents both from a small village of Toledo, Spain, with a population of approximately 4400 people. His family history was remarkable for EOAD in his father, starting at 61 years old with memory complaints (Figure 2A), who died around 80 with severe dementia. The paternal grandparent was also diagnosed with AD (onset unknown). The mother of the proband turned 80 years old cognitively intact, but then, mild behavioral impairment was noted, dying at the age of 92 with mild cognitive problems. He had two living siblings without cognitive impairment, aged 74 and 72, and one sister with an unspecified psychiatric disorder. Two brothers without cognitive difficulties died at 56 and 68 years old due to pancreatic adenocarcinoma and acute leukemia, respectively. His double cousin also developed EOAD, starting with memory problems before the age of 65. There is no available information about her parents.

The personal history of the proband was only remarkable for asymptomatic haemochromatosis. He carried the two most common *HFE* variants (p.C282Y and p.H63D) in compound heterozygosis. He was diagnosed as part of family members testing after the diagnosis of his symptomatic brother. Our proband had no systemic manifestations of iron overload.

Neurological examination on presentation showed moderate recent episodic memory impairment (Memory Impariment Screen (MIS) test score: 0/8) and marked constructive apraxia on bedside testing. He also exhibited marked difficulty on imitating meaningless gestures with both hands but performed well on pantomime of tool use. Mild calculation and executive deficits were also present as well as simultanagnosia. Language was scarce but formally preserved. There was neither parkinsonism nor oculomotor abnormalities. The remaining of the general and neurological examination was normal. He was independent for basic activities of daily living and required mild assistance for instrumental activities. Mini-Mental State Examination (MMSE) was 19/30. 

Global progression of cognitive impairment was observed during follow-up. Two years after presentation, he required moderate assistance for instrumental activities of daily living, and MMSE score was 11/30. Memory problems, constructive apraxia, gesture imitation and visuospatial deficits progressed significantly. He also developed marked anomia, impaired repetition of sentences, and mild comprehension problems. 

Magnetic resonance imaging (MRI) of the brain demonstrated moderate cortico-subcortical atrophy predominantly affecting both parietal lobes. Susceptibility-weighted images (SWI) showed superficial siderosis on the right frontal and occipital lobes, as well as multiple subcortical microhemorrhages, fulfilling modified Boston criteria for probable CAA (Figure 3).

Amyloid PET (18F-Florbetaben) was positive for cortical amyloid-β pathological deposition on qualitative analysis. He was diagnosed with dementia with an etiologically mixed presentation (concomitant AD and CAA) and was started on donepezil.

Genetic testing revealed the presence of the c.364C>T (p.R122*) variant in the *SORL1* gene in homozygosis. This variant has an extremely low allelic frequency (rs775517202), being reported only in one individual from the general population in the gnomAD database. It is located in exon 2 and it is predicted to truncate approximately 95% of the SorLA protein sequence. *APOE* genotype was 3/3.

### 2.2. Case 2

Patient 2 was a right-handed man who presented progressive cognitive decline starting from the age of 60. His family reported that he was forgetful and repetitive. He started to forget appointments or events, and he was usually disoriented in time. In addition, they described a mild behavioral impairment characterized by irritability and neglect of personal care. He was a regular cannabis, alcohol and tobacco user. Three years prior to presentation, he was diagnosed with a clear cell renal cell carcinoma, which was treated with a partial nephrectomy. He also had well controlled arterial hypertension. 

His family history was remarkable for early-onset dementia in his sister (Figure 2B). She was diagnosed with a probable AD at 46 years old and died 20 years later with severe dementia. His father died after suffering an ischemic stroke at the age of 50. His mother and brother also suffered a young-onset ischemic stroke. He also had a younger sister without any known neurologic comorbidities.

On the last neurological examination performed at the age of 63, the MMSE score was 20/30, losing points in orientation to time, delayed recall, sentence writing and figure copy. He exhibited moderate episodic memory impairment (MIS 2/8) and was disoriented in time. He was anosognosic about his deficits. Language, attention and visuospatial domains were preserved, and he required mild-to-moderate assistance for instrumental activities of daily living. He exhibited mild executive dysfunction, scoring 5/7 on a Clock-Drawing Test.

MRI of the brain performed on presentation showed moderate bilateral hippocampal atrophy and mild-to-moderate cerebral small vessel disease. 18F-FDG-PET demonstrated a left predominant temporoparietal hypometabolism. Amyloid-PET was positive for cortical amyloid-β deposition.

On the basis of profound amnestic deficits interfering with functional abilities, with relative preservation of other cognitive domains, he was clinically diagnosed with probable AD dementia and started treatment with rivastigmine. In addition, 18F-FDG-PET showed a pattern of neurodegeneration typical of AD, and pathological deposition of amyloid-β in a young patient increases the likelihood of AD as the etiology of this dementia syndrome. Nevertheless, the presence of mild-to-moderate cerebral small vessel disease might have contributed to the cognitive deficits observed.

Genetic testing revealed the novel pathogenic heterozygous c.4408C>T (p.R1470*) variant located in exon 32 of the *SORL1* gene. This variant is predicted to truncate approximately 34% of the SorLA protein sequence. Given the family history of young-onset stroke, we also specifically screened the *COL4A1*, *COL4A2*, *HTRA1*, and *NOTCH3* genes, discarding pathogenic variants. The *APOE* genotype was 3/3.

### 2.3. Case 3

Patient 3 was a 59-year-old man who referred a maternal history of LOAD. His mother and two aunts were diagnosed with AD dementia on the eighth decade of life. He did not have history of neurologic diseases in his paternal lineage (Figure 2C).

He complained of a 4-year history of mild episodic memory deficits that had worsened over the last year. He also described a subjective decrease in verbal fluency, which was barely noticeable on bedside testing. Additionally, during the last few months, he had become more anxious and distressed and noticed the worsening of a long-standing simple motor facial tic (repetitive eye blinking) that was treated in the past with botulinum toxin injections. These complaints did not affect his functional abilities.

Bedside examination showed mild-to-moderate deficits in verbal episodic memory of the hippocampal type. He was able to freely recall 3 out of 10 words after 5 min and did not improve with semantic clues. The remaining of cognitive domains were not impaired.

Brain MRI at 57 years old did not show significant atrophy or cerebral small vessel disease. He was diagnosed with amnestic mild cognitive impairment and received genetic testing due to patient desires, early age of symptoms onset, and family history of AD.

Genetic testing revealed the rare missense c.1906T>A (p.S636T) variant located in exon 14 from *SORL1* gene in heterozygosis. Its allelic frequency is low (rs138438079; 0.05%), and in silico software programs are contradictory on its prediction of pathogenicity (8 damaging and 10 benign). In addition, this patient was found also to carry the rare variant c.4033T>C (p.Cys1345Arg) in the *ABCA7* gene located in the disulfide bond domain of the protein. Its allelic frequency is extremely low (rs780608801; 0.0008%) and in silico software programs suggest to be deleterious. The *APOE* genotype was 3/4.

### 2.4. Case 4

Patient 4 developed a progressive cognitive impairment starting from 53 years old, dying bedridden twelve years later. He started presenting episodic memory impairment and mild anomic aphasia. Nevertheless, behavioral problems (apathy, irritability, impulsivity and binge eating disorder) were prominent since the beginning of cognitive deficits. On first examinations, he had marked impairment in learning and recalling new information. Word finding pauses due to anomic deficits were frequent as well as phonemic paraphasias.

Six years after the onset of symptoms, global cognitive impairment was evident, and he required complete assistance for basic activities of daily living. Irritability and aggressive behavior were a problem throughout the course of the disease. He also developed parkinsonian features in advanced stages, which were probably influenced by antipsychotic medication. 

His mother was diagnosed with EOAD dementia, with an age of onset before 60 (Figure 2D). He also had two brothers without cognitive impairment at the moment of the anamnesis. He had suffered a myocardial infarction at 49 years old, but he denied any other medical problems.

Brain MRI at the beginning of the clinical symptoms was normal, and SPECT tomography demonstrated frontal hypoperfusion. Another MRI performed at 56 years old showed mild-to-moderate global cortico-subcortical atrophy without significant small vessel disease.

Genetic testing revealed the novel heterozygous splicing variant c.4519+5G>A located in intron 32 of the *SORL1* gene. In silico tools suggest that this variant alters the 5′donor splicing site leading to exon skipping. *APOE* genotype was 3/3.

Diagnosis during life was possible AD dementia with an atypical course. Amnestic and language deficits dominated the clinical picture, but given the initial and prominent behavioral changes, behavioral variant frontotemporal dementia was included in the differential diagnosis throughout the course of the disease.

AD was confirmed after neuropathological assessment of the brain. Macroscopically, it was observed a diffuse cortical brain atrophy, predominantly affecting frontal and anterior temporal lobes, concurrent enlargement of ventricular system, and important atrophy of medial temporal lobe structures. Basal ganglia were preserved. In the brainstem, the locus coeruleus was pale and atrophic. The lateral part of substantia nigra was slightly pale.

Histopathologic assessment demonstrated changes compatible with the diagnosis of AD. ABC score was A3B3C2 (Thal 4, Braak & Braak V, CERAD 2), indicating high grade of AD neuropathological changes. Alpha-synuclein deposits were absent. Isolated neuronal cytoplasmic inclusions of transactive response DNA-binding protein (TDP-43) were observed in the hippocampus and amygdala, together with segmental hippocampal sclerosis. TDP-43 inclusions were not present in frontal lobes. Furthermore, the doctors observed leptomeningeal amyloid angiopathy (CAA-type 2) without capillary pathology (Figure 4). No inflammation or CAA-related angiitis was detected. There was no evidence of concomitant pathology of any other neurodegenerative disease.

In addition, the expression pattern of SorLA was characterized in the hippocampus from the patient and compared to those observed in a sporadic LOAD and one control without cognitive deficits. SorLA IHC revealed a finely granular cytoplasmic staining observed in pyramidal neurons at the somatodendritic compartment. This staining was slightly decreased in the patient, especially at the apical dendrites level. However, no clear differences were observed in terms of intensity and distribution when comparing neither to the control nor sporadic LOAD case (Figure 5).

### 2.5. Case 5

Patient 5 was a 69-year-old woman, diagnosed with amnestic mild cognitive impairment after a two-year history of insidious and progressive memory deficits with preservation of independence in functional abilities. MMSE score on presentation was 20/30, with deficits in delayed recall, orientation, calculation, and figure copy. MIS score was 1/8.

Slowly progressive global cognitive decline was observed during follow-up. One year after diagnosis, she required mild assistance for instrumental activities. At 71, ideomotor apraxia and memory deficits dominated the clinical picture. Language problems started at 72 years old, and four years after the first visit, a moderate global aphasic syndrome was patent. In parallel to language worsening, motor impairment with frequent falls and myoclonic jerks showed up in an advanced phase of the dementia. She died bedridden at the age of 75.

An MRI scan of the brain performed three years after the onset of symptoms showed mild global cortico-subcortical atrophy without small vessel disease. A cerebral SPECT performed in another institution demonstrated right temporal and posterior parietal hypoperfusion.

When she was 63, she had sought medical advice for mild apathy and depression. CT scan and cerebral SPECT were normal at that moment. She did not have any other comorbidities. Her father had been diagnosed with EOAD dementia, starting at the age of 55. Five of her father’s siblings also had LOAD, with an age of onset beyond 80 years old (Figure 2E).

This patient was diagnosed with probable AD dementia with documented clinical decline. Ancillary tests did not demonstrate any evidence of other pathologies that could explain the cognitive deficits. Although pathological amyloid-β was not tested, SPECT results showed a pattern typical of AD, and the phenotype, with initial and prominent episodic memory impairment, strongly suggested AD.

Genetic testing revealed the novel heterozygous missense c.5998G>A (p.G2000R) variant located in exon 44 from the *SORL1* gene. The prediction of pathogenicity according to the different software programs suggests that this variant is likely to be pathogenic. *APOE* genotype was 3/3.

### 2.6. Case 6

Patient 6 was a 53-year-old right-handed woman who presented with subacute and progressive language deficits. She started with word finding difficulties and mild phonemic paraphasias, which was followed by comprehension problems 6 months later. In addition, she described a mild amnestic syndrome and problems with tool use. 

She did not have family background of AD or other dementias (Figure 2F). She had two younger siblings cognitively unimpaired. The patient denied any other medical problems.

Neuropsychological assessment demonstrated mild memory deficits that did not improve with semantic clues, logopenic aphasia, and a complete Gerstmann syndrome, together with ideomotor apraxic problems, especially with the right hand. The MMSE score on presentation was 23/30. 

18F-Florbetapir amyloid-PET identified pathological cortical amyloid-β deposition, and MRI on presentation did not demonstrate significant atrophy. Brain SPECT scan showed mild asymmetric hypoperfusion affecting left temporal and frontal lobes. She was clinically diagnosed with the logopenic variant of primary progressive aphasia, fulfilling criteria for a probable AD dementia (language presentation), with evidence of AD pathophysiological process. 

Cognitive deficits slowly progressed, and 4 years later, her MMSE score was 1/30. Language impairment was the dominant symptom throughout the course of the disease. On advanced stages, the patient developed relevant behavioral symptoms (irritability, agitation, and physical aggression) and myoclonic jerks. She was institutionalized at 57 years old, and follow-up was lost.

Genetic testing was performed given the early age of symptoms onset. Researchers identified the rare missense c.6112C>G (p.H2038D) variant in heterozygosis in the *SORL1* gene. The allelic frequency of this variant (rs146761517) is very low (0.005%). It has been identified in only 12 individuals from the general population in the gnomAD database. This variant is located in exon 45 and pathogenicity software programs are discrepant on their prediction of pathogenicity. *APOE* genotype was 3/3.

## 3. Discussion

WES has proven to be a powerful tool in identifying causative pathogenic variants in known genes associated with neurodegenerative dementias [22]. The majority of AD cases present a multifactorial inheritance pattern where there are complex interactions between the collective effect of genetic factors that provide susceptibility to the disease combined with a variety of environmental factors that may trigger, accelerate, or protect against the disease-altered mechanisms [23]. From our 37 patients, WES identified disease-causing mutations in 8% (3/37) and likely pathogenic variants in dementia causal genes in 11% (4/37). In addition, 35% (13/37) of the studied cohort carried risk variants in genes associated with dementia (*APOE*, *SORL1*, *ABCA7* and *TREM2* genes). Among them, WES identified three men diagnosed with familial early-onset dementia (case 1, case 2 and case 3) carrying variants in the *SORL1* gene, which represented 15% (3/20) of all patients carrying candidate variants identified in our study. Two of them received a diagnosis of probable AD, and the other one had also evidence of concomitant CAA. Furthermore, we report on the clinical features of three additional patients (one man and two women) from the memory clinic of the Hospital 12 de Octubre diagnosed with AD, who also carried variants in the *SORL1* gene. Specifically, we report clinical, neuropathological, and IHC findings of a young male who also had a family history of EOAD (case 4). Regarding the two women, case 5 presented with a mixture of both EAOD and LOAD family history, while case 6 did not present family history of AD. These results reinforce the complex genetic etiology of dementias and in particular of AD and highlight the role of the *SORL1* gene, which is in line with previous reports [4,12,18]. Nevertheless, since genetic analysis was focused on 46 genes related to AD and dementia, we cannot exclude the presence of other genetic variants in other genes or regulatory regions that may be involved in the development of dementia.

To our knowledge, we report for the first time an AD patient with a homozygous nonsense variant (c.364C>T, p.R122*) in the *SORL1* gene. In the literature, there is a report of an EOAD patient with 55 years of age at onset with biallelic splicing variants in the *SORL1* gene [24]. This patient was found to carry a paternally inherited c.1211+2T>G variant and a complex allele c.[3947-3_3947-2insG;4265A>G] maternally inherited. Both variants were demonstrated to alter the splicing of the *SORL1* gene by ex vivo splicing assays using pCAS minigenes [24]. He had a family history of dementia in both maternal and paternal lineages with later ages of onset than the proband himself [24]. Contrary to this report, the familial history of our homozygous patient was only positive in the paternal lineage; with his father being also diagnosed with EOAD. After identifying the homozygous nonsense variant in the *SORL1* gene, we further investigated the cognitive skills of his mother. Although clinical evaluation was not performed in our hospital, the mother did not have cognitive problems until she turned 80 years old. At that time, their relatives described mild behavioral impairment (apathy, mild inattention, and irritability), dying at the age of 92 with preserved functional abilities. Regarding the age of onset, our *SORL1* homozygous patient was 62 years old at diagnosis, which was one year later than his “presumed” heterozygous father. This fact is in consonance with the previous suggestion made by Le Guennec et al. that the biallelic alteration of *SORL1* is not associated with a significant earlier age at onset than heterozygous *SORL1* variant carriers [24]. Considering this fact and the lack of remarkable clinical signs of cognitive decline in his mother, our results suggest that *SORL1* might be contributing to AD development as a risk factor gene rather than as a major autosomal dominant gene. Accordingly, patient 6 carried a rare missense variant in the *SORL1* gene, albeit segregation analysis was not performed due to its frequency being presumed to be inherited. The patient was diagnosed with EOAD at the age of 53 and did not show familial history of AD, supporting that neither LOF nor missense variants are per se causative variants responsible of AD.

On the other hand, it is noteworthy that the *SORL1* homozygous patient, beyond the clinical diagnosis of AD, fulfilled the modified Boston criteria for probable CAA [25]. In addition, the neuropathological findings of case 4 included CAA-type 2, with leptomeningeal amyloid deposition. As noted above, SorLA plays an important role in APP processing, thereby supporting a causal involvement of the *SORL1* gene. There is one case report of CAA-related inflammation in a patient with the rare missense variant c.4901A>T, p.K1634M in the *SORL1* gene [26]. However, this patient also had *APOE* ε4 homozygosity, which is known to be associated with an increased burden of amyloid-β deposition in blood vessels [27] and is strongly associated with CAA-related inflammation [28]. We did not detect any inflammation or CAA-related angiitis in the patient who underwent neuropathological evaluation. Louwersheimer et al. reported a family with genetic analysis of four members with AD and variable presence of microbleeds/CAA, who carried a rare variant of *SORL1* (NM.003105: c.2021A>G, p.Asn674Ser), but were also homozygous for the *APOE* ε4 allele and a mutation in the *TSHZ3* gene, which is also associated with amyloid-β processing [29]. In addition, CAA is described among the neuropathological findings of two patients with the c.3907C>T (p.Arg1303Cys) *SORL1* (NM.003105.5) variant who were heterozygous for *APOE* ε4 [30]. Our report suggests that *SORL1* might play a role in the development of CAA, even in the absence of an *APOE* ε4 allele. The finding of some SNPs of *SORL1* associated with an increased risk of microbleeds in a Dutch cohort of hypertensive individuals may support this causative role [31]. Nevertheless, it is important to remark on the high prevalence of moderate-to-severe CAA in sporadic AD, reaching almost 50% based on neuropathological examination [32].

Regarding the inheritance pattern, it cannot be discarded that *SORL1* variants may show incomplete penetrance or rather are based in a potential oligogenic combination modulated by other genetic factors such as *APOE* genotype. Insights from the report of Louwersheimer et al. suggest that rare genetic variants in the *SORL1* gene may increase the penetrance of homozygosity of the *APOE* 4/4 genotype in AD patients [29]. From our cohort, case 3, with 55 years of age at onset, was found to carry a rare missense variant in the *SORL1* gene together with one *APOE* ε4 allele and the rare variant rs780608801 in the *ABCA7* gene in heterozygosis. He had a family history of LOAD in the maternal linage. Nevertheless, we do not know whether these variants were inherited from the same parent or from both, since no genetic information from his mother or his maternal uncles was available. Although his clinical diagnosis was amnestic, mild cognitive impairment and amyloid-β deposition has not been demonstrated, yet the fact that he is carrying three risk variants makes his clinical follow-up interesting in order to see the impact of these variants in his clinical progression. Similarly, there has been reported an EOAD individual with onset in the mid-50s and a strong family history of AD who carried the missense *SORL1* variant p.M105T and one *APOE* ε4 allele [33]. Furthermore, Thonberg et al. reported three *SORL1* heterozygous variants in three families where affected individuals carried at least one *APOE* ε4 allele [30]. Nevertheless, most of the patients herein reported with *SORL1* variants showed the common *APOE* 3/3 genotype (5/6), indicating that there must be additional factors involved in the development of *SORL1*-associated AD. In this line, the presence of the p.Arg47His variant in the *TREM2* gene, as well as rare variants in the *ABCA7* gene, have been described as responsible for an increased risk of AD. In our cohort, only case 3 was found to carry a rare variant in the *ABCA7* gene, and the p.Arg47His was only found in one patient (3%; 1/37) who did not present any other causative or risk factor variant. On the basis of these observations and being aware of the limited sample size, we can speculate that *SORL1* has a greater impact in the risk of developing AD or early-onset dementia than p.Arg47His of the *TREM2* gene or rare *ABCA7* gene variants. The WES data generated from the patients herein reported might be screened in the future if new risk factor genes for AD are discovered.

This study has three limitations. Firstly, the lack of segregation analysis. However, it was not performed in asymptomatic relatives because predictive testing is not recommended without a genetic counseling consultation due to ethical reasons. Secondly, the lack of functional studies for *SORL1* variants. Nevertheless, post-mortem brain tissue was available from case 4, with the splicing variant c.4519+5G>A in *SORL1*. Neuropathological evaluation confirmed definitive AD diagnosis, and the IHC of SorLA revealed weaker expression in hippocampus, albeit no significant loss from those observed in sporadic AD and in the control. These results are in line with the atypical SorLA staining observed in two EOAD siblings (both heterozygous for *SORL1* variant and *APOE* ε4 allele), which was also found, albeit less pronounced, in sporadic AD cases and controls [30]. In addition, Barthelson et al. have recently reported on the effects on young-adult zebrafish whole brain transcriptome carrying similar mutations from the two nonsense (p.R122*, p.R1470*) herein reported [34,35]. These authors have generated in vivo models of a familial EOAD-like model carrying the p.V1482Afs variant in the orthologue *sorl1* gene, which mimics the human mutation p.C1478* and is likely subject to non-sense mediated decay (NMD) in peripheral blood [34]. In addition, the authors generated also the knock-in zebrafish model containing the null variant p.R122Pfs, as a control representing haploinsufficient LOF, since this mutation is predicted to encode a protein product containing only the 5% of the Sorl1 protein, lacking any of its functional domains. Gene set enrichment analysis from both types of mutants showed subtle functional consequences in brain transcriptome, evidencing changes in cellular processes in brain such as energy metabolism, protein translation, and degradation, which might be due to the haploinsufficiency of the *SORL1* mutation [34]. On the basis of these observations, we can speculate that the nonsense variants detected in case 1 and case 2 might lead to similar molecular consequences which could impact the normal function of the brain. Finally, the third limitation of this study is the lack of demonstration of amyloid-β pathology in two patients of this cohort (cases 3 and 5). Although biomarkers are not available, case 3 exhibits progressive amnestic deficits of the hippocampal type, and case 5 fulfills NIA-AA 2011 criteria for probable AD dementia with documented clinical decline. In addition, the SPECT pattern of hypoperfusion is typical of AD.

## 4. Materials and Methods

### 4.1. Individuals Included

Data from individuals reported in this study include two sets of patients. The first set consists in 37 patients with early-onset dementia or family history suggestive of autosomal dominant dementia (modified Goldman score 1) [36] who were referred from January 2018 to September 2020 to the Genetics Service of the Hospital Universitario 12 de Octubre (Madrid, Spain) for genetic testing. The second set consists of 3 AD patients with *SORL1* variants recruited from the memory clinic of Hospital 12 de Octubre. We retrospective reviewed 40 patients who underwent genetic testing between 2015 and 2018 in external laboratories (gene panels available under request).

The likelihood of dementia due to AD was established by applying the recommendations from the National Institute on Aging-Alzheimer’s Association workgroup (NIA-AA 2011 criteria) [2].

Written informed consent was obtained from all patients or their guardians prior to their participation, in accordance with institutional requirements and the Declaration of Helsinki Principles. The institutional review board approved the collection and use of these samples for research purposes (Ethics Committee of Hospital Universitario 12 de Octubre). DNA extraction was performed from peripheral blood following standard procedures.

### 4.2. NGS and Data Analysis

Whole-exome sequencing (WES) libraries were prepared using xGen Exome Panel v1.0 kit (Integrated DNA Technologies, Coralville, IA, USA) and paired-end sequencing (2 × 75 bp) was carried out on a NextSeq 550 sequencer (Illumina, San Diego, CA, USA). Whole-exome mean coverage at a sequencing depth ≥ 20x was 95.9 (95.4–96.4) and the mean depth of coverage was 85.4x (71.3–99.5). A validated custom pipeline -KarMa- was used for the Bioinformatics analysis. The pipeline was validated following the recommendations of the Association of Molecular Pathology [37]. Reads were aligned to the reference human genome (hg19) using BWA MEM (v0.7.17) [38] and Bowtie2 (v.2.4.1) [39]. The variant calling process was performed using Haplotype Caller from GATK (Genome Analysis Toolkit, v.4.1) [40] and VarDict (AstraZeneca, v1.7.0) [41]. Variation annotation was performed using ANNOVAR (v2018Apr16) [42].

WES data analysis was based on a custom panel that includes the following 46 genes related to dementia and AD: *ABCA7*, *ANG*, *APOE*, *APP*, *ATP13A2*, *ATXN2*, *C9orf72*, *CHCHD10*, *CHMP2B*, *CSF1R*, *DCTN1*, *DNMT1*, *FUS*, *GBA*, *GRN*, *HNRNPA1*, *HNRNPA2B1*, *HTRA1*, *ITM2B*, *LRRK2*, *MAPT*, *MATR3*, *NOTCH3*, *OPTN*, *PINK1*, *PRKAR1B*, *PRNP*, *PSEN1*, *PSEN2*, *SERPINI1*, *SIGMAR1*, *SNCA*, *SNCB*, *SORL1*, *SQSTM1*, *TARDBP*, *TBK1*, *TBP*, *TIMM8A*, *TOMM40*, *TREM2*, *TRPM7*, *TUBA4A*, *TYROBP*, *UBQLN2* and *VCP* genes.

Potential pathogenic variants for the genes included in the custom panel were filtered according to quality parameters, variant type, pathogenicity predictor scores, and variant frequencies in population control databases such as allelic frequencies in Genome Aggregation Database (v2.1.1) and frequencies in our in-house database of variants (12OVar). Pathogenicity scores include 14 prediction software for calculating impact of missense variants, 4 software for splicing alteration and 7 conservations scores. Variant classification was made following the ACMG criteria [43]. In addition, *APOE* genotype and p.R47H variant in *TREM2* gene were screened in all samples as they are considered risk factor variants.

### 4.3. Neuropathological Evaluation

AD neuropathology was evaluated in a post-mortem brain sample of one case diagnosed with EOAD who carried a pathogenic splicing variant in *SORL1* gene (see case 4 description). AD evaluation included the Braak and Braak stage, the Thal phases, and the Consortium to Establish a Registry for Alzheimer Disease (CERAD) plaque score, using the Diagnostic Criteria for the Neuropathological Assessment of Alzheimer’s Disease, following the recommendation by the National Institute on Aging-Alzheimer’s Association workgroup [44].

In addition, tissue from blocks of CA1 sector of the hippocampus at the level of the lateral geniculate body were obtained for immunohistochemistry (IHC) in case 4 and two additional samples: a patient diagnosed with LOAD and a 40-year-old control individual who died without neurological manifestations. Microscopic evaluation was performed on a 4 μm-thick sections of formalin fixed, paraffin-embedded material. Standard IHC was performed with a primary antibody against SorLA (sorting protein-related receptor with A-type repeats) following manufacture’s recommendations (Abcam, Cambridge, UK; ab190684, 1:400).

## 5. Conclusions

Our report might represent one of the largest and detailed clinical descriptions of *SORL1* patients, including MRI imaging, amyloid biomarkers, and one case with confirmed AD by neuropathological examination, where SorLA IHC was also performed. These series highlight the clinical heterogeneity and disparate familial dementia background associated with *SORL1* mutations. Furthermore, we report for the first time a homozygous case of *SORL1* truncating variant, and a case of CAA associated with *SORL1* mutation in the absence of the *APOE* ε4 allele. Altogether, our results support that although *SORL1* might not be fully associated with autosomal dominant inheritance, it might be considered a major dementia gene risk, especially in AD. Based on our results and in consonance with the current literature, rare missense or LOF variants should be reported because they may contribute to explain a large part of the etiology in the carriers. Nevertheless, further studies are required in order to shed light on the AD model of inheritance and the penetrance associated with *SORL1* genetic variants.

## Figures and Tables

**Figure 1 ijms-23-04230-f001:**
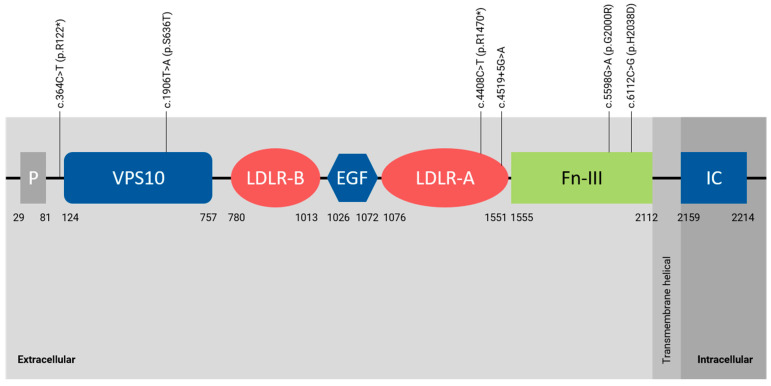
Graphical description of *SORL1* variants (NM_003105.6) identified in this report. Abbreviations. EGF: epidermal growth factor like domain; Fn-III: fibronectin-type III domain; IC: intracellular domain; LDLR-A and LDLR-B: low density lipoprotein receptor A and B domains; P: pro-peptide; VPS10: vacuolar protein sorting 10 domain.

**Figure 2 ijms-23-04230-f002:**
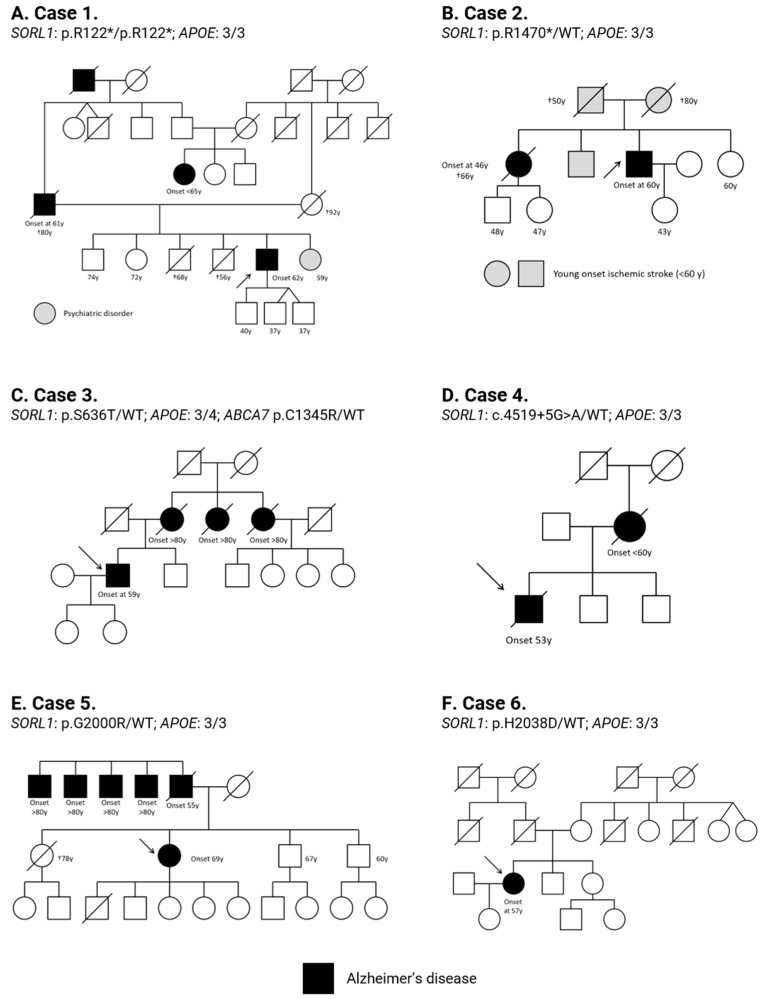
Familial pedigree of cases with *SORL1* variants reported in this study. (**A**) case 1; (**B**) case 2; (**C**) case 3; (**D**) case 4; (**E**) case 5; and (**F**) case 6. Arrows show probands; black-filled symbols represent AD affected individuals. Gray-filled symbols represent relatives affected with other neurological issues. Genotypes for *SORL1* and *APOE* genes are shown. WT: wild-type allele.

**Figure 3 ijms-23-04230-f003:**
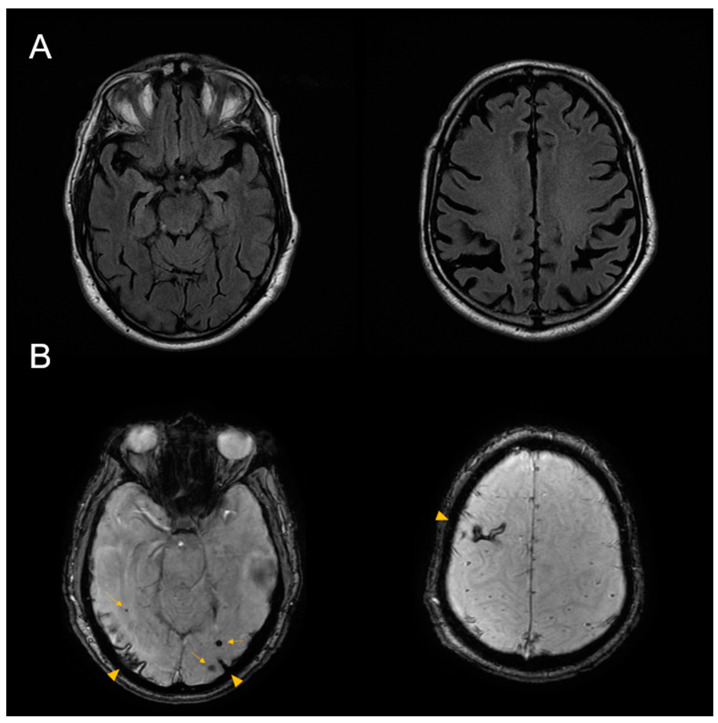
Brain MRI from case 1. (**A**) Axial fluid-attenuated inversion recovery (FLAIR) sequence shows global atrophy, predominantly involving both parietal lobes. (**B**) Susceptibility weighted imaging (SWI) sequence shows right frontal and bilateral occipital superficial siderosis (arrow heads). Multiple lobar microhemorrhages were also present (arrows).

**Figure 4 ijms-23-04230-f004:**
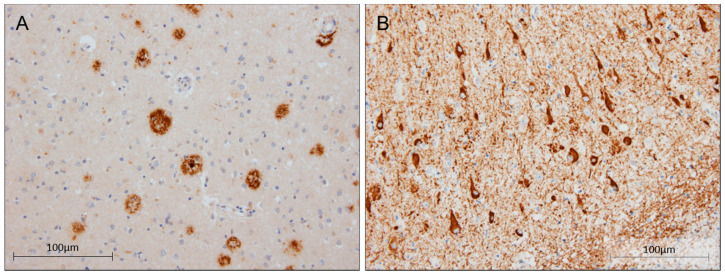
Post-mortem brain tissue from case 4. (**A**) Immunohistochemistry (IHC) staining of β-amyloid in parietal association cortex configuring amyloid plaques (20×). (**B**) Intense IHC staining of tau in the cortex of the middle temporal gyrus (20×). Scalebar: 100 μm.

**Figure 5 ijms-23-04230-f005:**
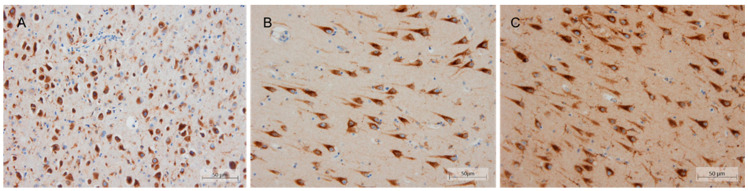
Immunohistochemical localization of SorLA in post-mortem brain tissue (20×). Representative pictures from (**A**) case 4, (**B**) sporadic AD, and (**C**) control in the CA1 hippocampal region using ab190684 *SORL1* antibody. Scalebar: 50 μm.

**Table 1 ijms-23-04230-t001:** Main clinical findings of Alzheimer’s disease patients with variants in the *SORL1* (NM_003105.6) gene.

Case ID	Age at Onset	Gender	Modified Goldman Score ^¥^	First Affected Domains ^†^	Imaging	Last Evaluation	Comments	Diagnosis	AD Family History	Genetic Findings
**Case 1**	59	Male	2	**Memory** Behavior Visuospatial Praxis	**MRI:** global atrophy, parietal predominant. Boston criteria for probable CAA. **Amyloid-PET:** positive.	65 yo. Moderate dementia. MMSE 11/30. Global deterioration and emergence of language symptoms.	Boston criteria for probable CAA. Stop variant in homozygosis in *SORL1*	Dementia with an etiologically mixed presentation (concomitant AD and CAA)	Paternal EOAD	*SORL1* c.364C>T (p.R122*) hom *APOE* 3/3 *HFE*: p.[C282Y][H63D]
**Case 2**	60	Male	3	**Memory** Behavior Executive	**MRI:** moderate SVD. Mild cortical and moderate hippocampal atrophy. **Amyloid-PET:** positive. **18F-FDG-PET:** left predominant temporoparietal hypometabolism.	63 yo. Mild dementia. MMSE 20/30. Mild memory impairment and irritability. Anosognosia.	Family history of young onset stroke. No pathogenic variants in *NOTCH3*	Probable AD dementia with high evidence of AD pathophysiological process	Parents no available. Sister EOAD	*SORL1* c.4408C>T (p.R1470*) het *APOE* 3/3
**Case 3**	55	Male	1	**Memory**	**MRI:** normal.	59 yo. Moderate hippocampal memory deficits. Anxiety.	Long-standing history of simple motor facial tics (repetitive eye-blinking)	Amnestic mild cognitive impairment	Maternal LOAD	*SORL1* c.1906T>A (p.S636T) het *APOE* 3/4 *ABCA7* c.4033T>C (p.C1345R) het
**Case 4**	53	Male	3	Memory Behavior Language	**MRI:** mild global atrophy. **SPECT:** right temporal and posterior parietal hypoperfusion.	Deceased at 65 yo with a severe dementia. Requirement of complete assistance at 59 yo.	High grade of AD neuropathological changes (A3B3C2). Marked behavioral problems	Possible AD dementia with an atypical course (during life). Pathophysiologically proved AD dementia (post-mortem)	Mother with EOAD	*SORL1* c.4519+5G>A het *APOE* 3/3
**Case 5**	67	Female	1	**Memory**	**MRI:** normal. **Amyloid-PET:** positive. **SPECT:** mild left asymmetric hypoperfusion.	Deceased at 75 yo with a severe dementia.	Mild apathy and depression 4 years before the onset of cognitive symptoms	Probable AD dementia with documented decline	Father with EOAD and paternal LOAD	*SORL1* c.5998G>A (p.G2000R) het *APOE* 3/3
**Case 6**	53	Female	4	Memory **Language** Praxis	**MRI:** normal. **Amyloid-PET:** positive. **SPECT:** mild left asymmetric hypoperfusion.	Follow-up lost at 57 yo, with a severe dementia (MMSE 1/30)	Subacute onset. Prominent language symptoms starting as lvPPA	AD dementia (language presentation, logopenic variant) with evidence of AD pathophysiological process	No family history	*SORL1* c.6112C>G (p.H2038D) het *APOE* 3/3

^¥^ Modified Goldman scores were obtained through analysis of family history data and separated into 5 categories: (1) autosomal dominant: 3 affected individuals over 2 generations with 1 person being a first-degree relative of the other 2; (2) familial aggregation: 3 relatives affected without satisfying the criteria for autosomal dominant inheritance; (3) single affected first-degree relative younger than the age of 65 years, (3.5) single affected first-degree relative older than the age of 65 years old; and (4) a family history that does not satisfy the previous classifications or no family history. ^†^ The most affected domain is highlighted in bold. Abbreviations. AD: Alzheimer’s disease; CAA: cerebral amyloid angiopathy; EOAD: early-onset Alzheimer’s disease; het: heterocygote; hom: homocygote; LOAD: late-onset Alzheimer’s disease; lvPPA: logopenic variant of primary progressive aphasia; MMSE: mini-mental state examination; MRI: magnetic resonance imaging; SPECT: single-photon emission computerized tomography; SVD: small vessel disease; yo: years old.

**Table 2 ijms-23-04230-t002:** Molecular description of *SORL1* variants (NM_003105.6) identified in this report.

Case ID	cDNA	Protein Change	Zygosity	Location	Frequency gnomAD	dbSNP ID	ACMG Classification	*APOE* Genotype
**Case 1**	c.364C>T	p.R122*	HOM	Exon 2	0.0004%	rs775517202	LP	3/3
**Case 2**	c.4408C>T	p.R1470*	HET	Exon 32	NF	-	LP	3/3
**Case 3**	c.1906T>A	p.S636T	HET	Exon 14	0.05%	rs138438079	VUS	3/4
**Case 4** ^§^	c.4519+5G>A	p.?	HET	Intron 32	NF	-	LP	3/3
**Case 5**	c.5998G>A	p.G2000R	HET	Exon 44	NF	-	VUS	3/3
**Case 6**	c.6112C>G	p.H2038D	HET	Exon 45	0.005%	rs146761517	VUS	3/3

^§^ This patient has already been reported in [21]. Abbreviations. ACMG: American College of Medical Genetics; HET: heterozygous; HOM: homozygous; LP: likely pathogenic; NF: not found; VUS: variant of unknown significance.

## Data Availability

All sequencing data from this study is available in the Genetic Service of the Hospital 12 de Octubre, under request.

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
