# Peer review of "Heterozygous and Homozygous Variants in SORL1 Gene in Alzheimer’s Disease Patients: Clinical, Neuroimaging and Neuropathological Findings"

_ijms, 2022, doi:10.3390/ijms23084230_

Round 1

Reviewer 1 Report

In this manuscript, Alvarez-Mora et al., report their genetic and clinical findings following WES of 37 EOAD patients.  Specifically targeting SORL1 variants, they report three cases with heterozygous or homozygous variants, plus an additional three cases from retrospective review of prior sequence data of patients from the Hospital 12 de Octubre. 

The authors have conducted a thorough clinical and neuropathological analysis, plus collected familial/pedigree information for each of the 6 SORL1 variant cases, and in general this is a well-written case-report, that will be of general interest to the scientific community.  There are a couple of minor suggestions for clarification.

  • The authors should clearly state in the abstract that WES was performed on 47 candidate genes. The discussion should include a sentence to state that they can not exclude the possibility of variants in regulatory regions and/or other genes.   
  • The number of subjects employed in the retrospective review from the Hospital 12 de Octubre should be documented in the results and methods.
  • The authors should specify the allele frequency for all variants (some are noted, for example case#3, but others are not, such as case#1).
  • It would be interesting for the reader if the authors report the actual contradictory findings of the in-silico prediction for case#3.
  • In the discussion, the statement on lines 448-450 is somewhat premature, given the very small number of cases reported here, and should be amended accordingly.
  • In line 464, the authors may have meant to write “his” instead of “her”.
  • The IHC data does not provide much benefit to the clinical or genetic data, and could be removed (along with the extensive discussion of this date and models systems); especially if the length of the case report was a review factor.

Author Response

Reviewer 1.

In this manuscript, Alvarez-Mora et al., report their genetic and clinical findings following WES of 37 EOAD patients. Specifically targeting SORL1 variants, they report three cases with heterozygous or homozygous variants, plus an additional three cases from retrospective review of prior sequence data of patients from the Hospital 12 de Octubre.

The authors have conducted a thorough clinical and neuropathological analysis, plus collected familial/pedigree information for each of the 6 SORL1 variant cases, and in general this is a well-written case-report, that will be of general interest to the scientific community. There are a couple of minor suggestions for clarification.

The authors should clearly state in the abstract that WES was performed on 47 candidate genes. The discussion should include a sentence to state that they can not exclude the possibility of variants in regulatory regions and/or other genes.

We would like to thank the reviewers for their appraisal review of the manuscript. He have modified the abstract by adding the following sentence:

“Data analysis was based on a custom panel that included 46 genes related to AD and dementia.”

In addition, we have added the following sentece in the first paragraph of the discussion:

Nevertheless, since genetic analysis was focused on 46 genes related to AD and dementia we cannot exclude the presence of other genetic variants in other genes or regulatory regions that may be involved in the development of dementia.

The number of subjects employed in the retrospective review from the Hospital 12 de Octubre should be documented in the results and methods.

Before the implementation of whole exome sequencing in the Genetic Service of the Hospital 12 de Octubre, there were not specific guidelines for performing genetic tests to patients with dementia in our hopsital.

The reteospective review encompasses genetic studies performed between 2015 and 2018 to 40 patients. As suggested by the reviwer, he number of patients has been included in the material and methods section.

The authors should specify the allele frequency for all variants (some are noted, for example case#3, but others are not, such as case#1).

Allelic frequencies of SORL1 variants are shown in Table 2. In addition, in the results               section, the allelic frequency of those variants which are present in public data bases               are also noted (case 1, 3 and 6). Finally, as described in the manuscript variants               detected in cases 2,4 and 5 are               novel.

However if the reviewer considers that this data should appear before in the               manscucript we can modify it.

It would be interesting for the reader if the authors report the actual contradictory findings of the in-silico prediction for case#3.

Using varsome (rs138438079 SNV | hg38 (varsome.com)), the variant detected in case               3 is predicted to be damaging by 8 in silico tools and benign by 10 predictor tools. This               information has been added in the mansucript.

In the discussion, the statement on lines 448-450 is somewhat premature, given the very small number of cases reported here, and should be amended accordingly.

We have modified this sentence in order to make its meaning less taxative. Given that               SORL1 gene is involved in the amyloid pathway, and in light of the clinical findings, we               state that it might play a role, instead of highlighting its relevance.

Our report suggests that SORL1 might play a role in the development of CAA.”

In line 464, the authors may have meant to write “his” instead of “her”.

We have corrected this mistake.

The IHC data does not provide much benefit to the clinical or genetic data, and could be removed (along with the extensive discussion of this date and models systems); especially if the length of the case report was a review factor.

Thank you for pointing this out. We have trimmed the extensive discussion of the               functional consequences of SorLa protein insufficiency addressed in zebrafish models               by Barthelson et al, and removed the reference 36. Nevertheless, we have kept IHC data               for two reasons: first, it is in line with the only report in the literature that evaluated               this fact; and second, it highlights that hippocampal SorLa expression might not be of               relevance in the development of AD.

Reviewer 2 Report

Authors presented the extensive study of AD patients by WES with many institutions and are reporting significance with SORL1 mutations among the group. Authors also correlated the SORL1 mutations to clinical symptoms, neuroimages and neuropathological observations. They further investigated the family members of patients with SORL1 mutations.

It would be interesting to further support their genetic results with other minor risk factor genes and biomarker study, especially with plasma biomarkers Ab42, tTau, pTau, NfL and Ab oligomers on non-symptomatic family members for the AD prevention, which would require the genetic counselings.

English grammar needs to be improved as following, run-on sentences and verb tenses

Author Response

Reviewer 2.

Authors presented the extensive study of AD patients by WES with many institutions and are reporting significance with SORL1 mutations among the group. Authors also correlated the SORL1 mutations to clinical symptoms, neuroimages and neuropathological observations. They further investigated the family members of patients with SORL1 mutations.

It would be interesting to further support their genetic results with other minor risk factor genes and biomarker study, especially with plasma biomarkers Ab42, tTau, pTau, NfL and Ab oligomers on non-symptomatic family members for the AD prevention, which would require the genetic counselings.

Thank you for your review of the manuscript. This point is very interesting. Amyloid status, as well               as CSF and plasma biomarkers of non-symptomatic SORL1 mutations might be of utmost impor              tance in the assessment of this gene risk of AD. As you point out, genetic counseling is required to               study asymptomatic relatives due to ethical reasons. We will consider this suggestion in future               works.

English grammar needs to be improved as following, run-on sentences and verb tenses

We have double-checked the english grammar, corrected some mistakes and run-on sentences
